# Probiotic Properties and Proteomic Analysis of *Pediococcus pentosaceus* 1101

**DOI:** 10.3390/foods12010046

**Published:** 2022-12-22

**Authors:** Monserrat Escobar-Sánchez, Ulises Carrasco-Navarro, Carmen Juárez-Castelán, Luis Lozano-Aguirre Beltrán, M. Lourdes Pérez-Chabela, Edith Ponce-Alquicira

**Affiliations:** 1Departamento de Biotecnología, Universidad Autónoma Metropolitana Unidad Iztapalapa, Ciudad de México 09340, Mexico; 2Cinvestav, Departamento de Genética y Biología Molecular, Ciudad de México 07360, Mexico; 3Unidad de Análisis Bioinformáticos del Centro de Ciencias Genómicas, UNAM, Cuernavaca 62210, Mexico

**Keywords:** proteomic analysis, antimicrobial activity, *Pediococcus pentosaceus*, probiotic

## Abstract

*Pediococcus pentosaceus* 1101 was identified by using 16S rRNA and MALDI-Biotyper. The strain was exposed to conditions that resemble the gastrointestinal tract (GT) to evaluate its probiotic properties. That included the growth kinetics, proteolytic and inhibitory activities within a pH range, survival at low pH and in the presence of bile salts, antagonistic activity, cell-adhesion properties, and antibiotic resistance. The evaluation was followed by a genomic and proteomic analysis that involved the identification of proteins obtained under control and gastrointestinal conditions. The strain showed antagonistic activity against Gram-negative and Gram-positive bacteria, high resistance to acidity (87% logarithmic survival rate, pH 2) and bile salts (99% logarithmic survival rate, 0.5% *w*/*v*), and hydrophobic binding, as well as sensitivity to penicillin, amoxicillin, and chloramphenicol. On the other hand, *P. pentosaceus* 1101 has a genome size of 1.76 Mbp, with 1754 coding sequences, 55 rRNAs, and 33 tRNAs. The proteomic analysis showed that 120 proteins were involved in mechanisms in which the strain senses the effects of acid and bile salts. Moreover, the strain produces at least one lytic enzyme (*N*-acetylmuramoyl-L-alanine amidase; 32 kDa) that may be related to the antimicrobial activity. Therefore, proteins identified might be a key factor when it comes to the adaptation of *P. pentosaceus* 1101 into the GT and associated with its technological and probiotic properties.

## 1. Introduction

At present, probiotics have been used in the production of several functional foods, dietary supplements, and biopharmaceuticals [1]. The term “probiotic” refers to a culture of live microorganisms that, when administered in adequate amounts, confer a benefit to the health of the host, such as immunomodulatory, anticancer, lipid-lowering, and antagonistic effects on pathogenic bacteria, that reinforce the intestinal barrier and balance the intestinal microbiota [2,3]. Most probiotics in humans belong to the lactic acid bacteria group (LAB) [4] that is commonly employed to produce several fermented food products, including dairy, meats, and vegetables. Probiotic strains comprise members of the genera *Pediococcus*, *Latilactobacillus*, *Bifidobacterium*, and *Enterococcus*, among others, and are generally recognized as safe [5].

It has been established that microbial isolates intended to be used as probiotics must meet several criteria, such as their ability to survive through the upper gastrointestinal (GI) tract and tolerate human gastric juices, as well as human bile and pancreatic secretion during transit through the intestine. Probiotic bacteria must also show adherence to epithelial surfaces to survive in the host GI and exert their beneficial activities on it [6]. Colonization of the gut by probiotic bacteria may prevent the growth of harmful bacteria by competitive exclusion, modulation of the immune system, and the production of organic acids among other antimicrobial compounds [7]. LAB are normal residents of the GI. The number of LAB in the stomach is <3 log CFU/mL; in the ileum, 2–5 log CFU/g; and in the colon, 4–9 log CFU/g [8]. 

LAB with probiotic properties can also be used as starter cultures in probiotic food products such as fermented meat products, thus contributing to the development of texture, taste, and shelf life. Probiotic LAB must be adapted to the heterogeneous environment found in meat and other food systems; they also must be a strong competitor against the natural microflora and grow in a number that has a beneficial effect on health [9]. The antagonistic activity of LAB is mainly due to the production of organic acids derived from the glucose metabolism, but other possible metabolites include bacteriocins and peptidoglycan hydrolases (PGH), among other antimicrobial peptides, which are also responsible for antimicrobial activity against pathogenic and spoilage microorganisms [10,11]. The cellular response to extreme environmental variations, or responses to stress, is characterized by changes in gene expression and cell physiology. These changes allow microorganisms to adapt to new environmental conditions [12], where proteases and PGH can be constitutive or partially inducible, and their synthesis is strongly influenced by factors such as the medium components, pH, and temperature variations. 

The genus *Pediococcus* belongs to the group of LAB, phylum Firmicutes, class *Bacilli*, order *Lactobacillales*, and family *Lactobacillaceae* [13]. The genus *Pediococcus* spp. includes several species, such as *P. acidilactici*, *P. pentosaceus*, *P. inopinatus*, *P. dextrinicus*, *P. clausseni*, *P. cellicola*, *P. ethanolidurans*, *P. parvulus*, and *P. stilesii*. In particular, *P. pentosaceus* has been reported to be part of the microbiota of several foods, such as cheese, beverages, pickles, wine, dairy products, and meat [14]. It is commonly used as a starter culture due to its acidifying and proteolytic activity, which influences the sensory characteristics of several fermented products, such as fermented meats [15]. In addition to their potential use as starter cultures, different strains belonging to the genus *Pediococcus* have potential use as probiotic bacteria [5], exerting beneficial effects through a variety of complementary mechanisms, including resistance to acidity and bile salts and antimicrobial activity against pathogens [14]. Proteomic methos based on LC–MS/MS offer advantages in speed and reliability compared to other traditional methods and allows for the possibility to identify and characterize the global bacteria proteome, including the search of factors associated with pathogenicity, as well as with antibiotic sensibility [16]. Some reports related to the proteomic analysis of *P. pentosaceus* have shown that this bacteria is capable of modulating its proteome when exposed to different stressful environments such as heat, cold, acid, bile, and oxidative stress; in addition, some proteins have been identified and could be used as markers to assess the stress tolerance, and that may help to understand the stress tolerance mechanisms, thus providing new perspectives for the production of enhanced probiotics [17,18]. The strain of *P. pentosaceus* 1101 was identified and selected for evaluation due to its wide inhibition spectrum, high resistance to acidic conditions, and prevalence throughout the maturation process of a fermented meat product (Spanish-type chorizo) from which it was previously isolated by our group [19]. Therefore, our objective was to evaluate the probiotic characteristics of the *P. pentosaceus* 1101 strain and to identify those proteins involved in the resistance and adaptation to acidity and bile salts in order to obtain information on the mechanisms that regulate the survival of this strain into the conditions of the gastrointestinal tract for a possible probiotic use, for which we decided to carry out a proteomic analysis, using the label-free technique. In addition, its inhibitory capacity and proteolytic activity at different pH values were evaluated since these conditions can induce changes in the cell metabolism.

## 2. Materials and Methods

### 2.1. Strain Propagation

The strain *P. pentosaceus* 1101 was previously isolated by our group from fermented meat products (Spanish-type chorizo) [19]. Bacterial stocks were prepared through incubation with MRS broth (Man-Rogosa-Sharpe, BD Difco, USA; 37 °C for 24 h) [20], and then 50% glycerol was added; the stocks were then sealed and preserved at −80 °C. Prior to the analysis, the studied strain was reactivated in CGB medium (Casein Glucose Broth, BD Bioxon, Cuautitlán Izcalli, Mex., Mexico) and incubated for 12 h at 37 °C, under static conditions. 

On the other hand, all reagents were acquired from Sigma-Aldrich (St. Louis, MO, USA), unless specified otherwise.

### 2.2. Bacterial Identification

The identity of the *P. pentosaceus* 1101 strain was confirmed by using 16S rRNA gene sequencing. DNA was extracted from a centrifuged pellet of an 18 h culture, using the Wizard Genomic DNA purification kit (Promega, Madison, WI, USA), following the manufacturer’s specifications. The integrity of the extracted DNA was verified by electrophoresis, using a 0.8% (*w*/*v*) agarose gel, TAE buffer 1X (Tris-acetate 10 mM, EDTA 1 mM, pH 8.0). The 16S rRNA gene was amplified by using the PCR technique with the enzyme Taq DNA polymerase (Thermo Scientific, Waltham, MA, USA) and universal primers for the 16S rRNA gene (27f-5′GTT TGA TCM TGG CTC AG 3′ and 1492R-5′TAC GGY TAC CTT GTT ACG ACT T 3′). The employed PCR conditions were initial denaturation (1 cycle of 94 °C for 2 min), denaturalization (30 cycles of 94 °C for 0.5 min), alignment (30 cycles at 49 °C for 0.5 min), extension (30 cycles of 72 °C for 1 min), and final extension (1 cycle of 72 °C for 1 min). Conditions were programmed in a thermocycler EP-Gradient (Eppendorf, Mastercycler EP, San Diego, CA, USA). The PCR product was purified and sequenced by Macrogen Inc. (Seoul, Republic of Korea). The obtained sequences were analyzed by Basic Local Alignment Search Tool (BLAST) and compared with the Refseq database at the National Center for Biotechnology Information (NCBI).

The strain identity was corroborated by MALDI-Biotyper at the Divisional Mass Spectrometry Laboratory of the Universidad Autónoma Metropolitana by using the ethanol and formic acid extraction method [21]. Individual fresh colonies isolated from MRS–agar plates were placed into 1.5 mL vials and suspended in 300 μL of deionized water, followed by the addition of 900 μL of ethanol, 70% formic acid, and 1 μL of acetonitrile, sequentially, for the inactivation, rupture, and precipitation of the cell membrane; the plates were then centrifuged (12,096× *g*) for 2 min. After centrifugation, 1 μL of the supernatant was placed on a steel plate (Bruker, Leipzig, Germany) with 1 μL of the matrix (α-ciano-4-hydroxycinnamic acid; Sigma-Aldrich, St. Louis, MA, USA). The analysis was performed on a MALDI–TOF-MS Autoflex speed mass spectrometer (Bruker Daltonics, Leipzig, Germany), using the MALDI Biotyper tool, and the data were compared to the MBT Compass database to assign genera and species. 

### 2.3. Genome Sequencing Assembly

The genomic DNA was extracted by using the RiboPure Kit (Invitrogen, Thermo Fisher Scientific, Waltham, MA, USA). Genome sequencing was performed by using the Illumina NextSeq platform, and a total of 5,326,286 paired reads (2 × 150) were obtained. Read quality control was assessed with FastQC software v0.11.9 (https://www.bioinformatics.babraham.ac.uk/projects/fastqc/, accessed on 2 November 2022), and quality and adapter trimming were performed with Trim Galore v0.6.7 (https://www.bioinformatics.babraham.ac.uk/projects/trim_galore/, accessed on 2 November 2022). De novo genome assembly was carried out with the SPAdes assembler v3.14.1 [22]. The *Pediococcus pentosaceus* 1101 genome was annotated by using the Prokka pipeline v1.14.6 [23].

### 2.4. Evaluation of Probiotic Properties

The probiotic parameters involve the evaluation of the antagonistic activity and its growth kinetics, proteolytic and inhibitory activities within a pH range; its survival at low pH and in the presence of bile salts that resemble the gastrointestinal tract; and the binding properties and its antibiotic resistance.

#### 2.4.1. Antagonistic Activity

The antagonistic activity of the strain was determined according to the method described by Juárez-Castelán [19] against Gram-negative and Gram-positive microorganisms *(Escherichia coli* DH5α GenBank: NZ_JRYM00000000.1, *Salmonella enterica* subsp. *enterica* serovar Typhimurium ATCC 14028, *Enterococcus faecalis* 1351, *Listeria monocytogenes* 1639, *Staphylococcus aureus* ATCC 6538, and *Weisella viridescens* GenBank: UAM-MG5: MT814884), which were previously activated at 37 °C in TSB (Tryptyc-Soy-Broth, Difco Laboratories, MA, USA) for 24 h. The antagonistic activity was performed in plates with agar–CGB inoculated vertically with the studied strain 1101 with a sterile swab and incubated at 37 °C for 24 h. Then the test microorganisms were inoculated perpendicular to the studied strain with a sterile swab, without touching, at approximately 3 mm distance from the vertical growth of strain 1101. Plates were incubated at 37 °C for 24 h. The positive reaction was indicated by the inhibition area close to the growth for each tested strain. 

#### 2.4.2. Growth Kinetics at Different pH Values

The kinetic parameters of strain 1101 were determined by measuring microbial growth under different pH conditions (2, 3, 4.5, 5, 5.5, 6, 6.5, and 7) at a constant temperature of 37 °C. *P. pentosaceus* 1101 cultivars from 24 h to 37 °C in MRS broth were prepared in tubes inoculated at 1% with the studied strain, and the optical density was adjusted to an A_600nm_ of 0.8. A volume of 200 μL was deposited on sterile microplates for each treatment and control, and the absorbance was recorded at 600 nm at time intervals of 30 min during incubation, with stirring, in a Synergy HT microplate reader (BioTek Instruments Inc., Winooski, VT, USA). Data were collected by using the Gen 5.11 software (BioTek Instruments Inc., Winooski, VT, USA). The obtained growth data were subjected to a Verhulst–Pearl logistic model [24] with the STATISTICA 7 program [25], and the kinetic parameters of growth rate (μ) and O.D.max were obtained with the following equation: (1)$$O.D.\left(t\right)=\left[\frac{O.D.\max}{1-\left(\frac{O.D.\max-{O.D.}_{0}}{{O.D.}_{0}}\right)\cdot e^{\left(-\mu t\right)}}\right]$$
where O.D._0_ refers to the value of O.D. at time t = 0. The minimum values of the square error were found as a function of the parameters O.D._0_, O.D.max, and μ.

Statistical analysis was performed by using NCSS 2007 software [26].

#### 2.4.3. Proteolytic Activity at Different pH Values

The skim-milk–agar plate technique [27] evaluated the strain’s proteolytic activity, using plates with 1.5% agar (BD, Bioxon, Cuautitlán Izcalli, Mex., Mexico) and 1.5% skim milk (Svelty, Nestle, Lagos de Moreno Jal., Mexico) that were prepared and sterilized separately at 121 °C for 15 min and 5 min, respectively, and combined for plate casting. Once the medium solidified, a series of wells were made with a sterile Pasteur pipette, 50 μL of each culture treatment was placed in them (pH 4.5, 5, 5.5, 6, 6.5, and 7), and the plates were incubated at 37 °C for 24 h. The presence of a halo around the well indicated positive protease activity indicative of protein degradation, and activity was reported as the diameter in millimeters of the halo.

#### 2.4.4. Inhibitory Activity at Different pH Values

The inhibitory-activity test was performed by using agar diffusion assays, as described by García-Cano [28], but with some modifications. Plates containing 15 mL of CGB with 1.5% agar were allowed to solidify at room temperature for 30 min, and then the plates were overlaid with 20 mL of 0.8% CGB agar previously inoculated with 140 μL (A_600nm_ of 0.3) of the indicator microorganism grown overnight in TSB under static conditions for 18 h at 37 °C. The indicator microorganism was *Listeria innocua* ATCC 33090. After media solidification, wells were made on the second layer of agar, using the posterior part of a 1000 μL micropipette tip. Then the inhibitory activity of *P. pentosaceus* 1101 cell-free extract was evaluated by placing 100 μL of the cell-free supernatant from each culture cell treatment (pH 4.5, 5, 5.5, 6, 6.5, and 7) into wells made in plates previously inoculated with *L. innocua* ATCC 33090. After incubation, the presence of a growth inhibition halo around the well indicated positive antimicrobial activity and was reported as the ratio of the diameter of the inhibition per mg protein. The protein concentration of the cell supernatant obtained from each treatment was determined by using the Bradford method [29], i.e., by mixing in a microplate 160 μL sample and 40 μL of Bradford reagent (Bio-Rad, Hercules, CA, USA) and measuring the absorbance at 595 nm after 5 min of incubation, and interpolated to a standard curve to determine the amount of protein.

#### 2.4.5. *P. pentosaceus* 1101 Survival after Exposure to Low pH and Bile Salts

Low-pH tolerance was assessed from a 15 h culture with an absorbance at 600 nm of 1–1.2 (10^9^ CFU/mL) of *P. pentosaceus* 1101 cells during the exponential growth phase. A 10% inoculum was taken for MRS media adjusted to pH 2 and 3 and incubated for 1 h at 37 °C, and the technique described by Khan [30] was used to determine cell viability (log_10_ CFU/mL) by agar–MRS plate counting. A 100 μL sample was taken from the tube and deposited in a Petri dish, using the surface-spreading technique in a plate with sterile glass beads, and an initial and final plate count of the culture was performed. The logarithmic survival rate was calculated by using the following equation:(2)$$\mathrm{Logarithmic}\;\mathrm{survival}\;\mathrm{rate}\;\left(\%\right)=\frac{\log10\;\mathrm{CFU}\;N1}{\log10\;\mathrm{CFU}\;N0}*100$$
where N1 represents the total viable count after incubation with the acidic treatment, while N0 represents the initial total viable count [7].

The ability of bacteria to grow in the presence of bile salts (bile extract porcine, Sigma-Aldrich, St. Louis, MA, USA) was determined from a 15 h culture of studied strain 1101 when it reached its exponential phase with an absorbance at 600 nm of 1–1.2 (10^9^ CFU/mL), and 10% inoculum was taken for the tubes with MRS adjusted with 0.5% swine bile (*w*/*v*). The same protocol described for acidity was used to determine cell viability; however, the tubes were inoculated at 10% with MRS adjusted with 0.5% bile salts and incubated for 6 h at 37 °C.

#### 2.4.6. Antibiotic Resistance

The antibiotic resistance profile against fourteen antibiotics was determined by using Clairo Combi Discs for Gram-positive bacteria (Arkray Healthcare, Surat, India) according to supplier recommendations. A bacterial suspension of *P. pentosaceus* 1101 corresponding to McFarland scale standard 1 was used and inoculated on Mueller–Hinton agar plates. The antibiotic sensitivity pattern was then tested by using the agar diffusion method, using the fourteen different antibiotics discs. After incubation at 37 °C for 24 h, the inhibition zones were measured inclusive of the diameter of the discs (7 mm). Results were expressed as sensitive, S (≥21 mm); intermediate sensitive, I (16–20 mm); and resistant, R (≤15 mm) [31].

#### 2.4.7. Cell-Adhesion Properties

For all bacterial adhesion tests, *Lactobacillus acidophilus* NCFM^®^ provided by Danisco USA Inc. (CA, USA) was used as a positive control, which involved cell-surface hydrophobicity, auto-aggregation, and co-aggregation capacities, as they have been correlated with the adhesion ability to epithelial surfaces that increases their potential colonization of the GT [32] and can be employed in a preliminary screening of potential probiotic bacteria [33].

Bacterial adhesion to solvents as an indicator of cell surface properties was performed as described by Sánchez [32], as cited in Hernández-Alcántara [5]. *P. pentosaceus* 1101 cells from a 15 h culture in the exponential phase (DO_560nm_ = 1–1.2; 10^9^ UFC/mL) were recovered by centrifugation (5000× *g* for 30 min) and washed twice with PBS (pH 7.2). The resultant pellet was resuspended to obtain a A_560nm_ = 0.6–0.7, mixed (*v*/*v*) with an organic solvent (xylene or chloroform), and then it was shaken vigorously, using a vortex, for 30 s. After 1 h of incubation at room temperature, the aqueous phase was carefully removed, and absorbance was measured at 560 nm (A_560nm_). The hydrophobicity of *P. pentosaceus* cells was calculated as follows: H [%] = [(A_o_ − A)/A_o_] × 100 (3)
where A_o_ and A correspond to the absorbance before and after extraction with organic solvents, respectively. 

The auto-aggregation capacity of *P. pentosaceus* was determined as described by Collado [33], with modifications made by Hernández-Alcántara [5]. Bacterial cells were recovered by centrifugation and washed twice with PBS (pH 7.2). They were then resuspended in the same buffer at A_600nm_ = 0.50 ± 0.10 to standardize the initial bacterial number (10^7^–10^8^ CFU/mL). The bacterial suspension was then incubated at room temperature and monitored at different time intervals (0, 2, 4, 6, 20, and 24 h). The percentage of auto-aggregation was expressed as follows:A [%] = [(A_o_ − A_t_)/A_o_] × 100 (4)
where A_o_ represents the absorbance at 0 h, and A_t_ represents the absorbance at time interval, t.

For co-aggregation assays, bacterial suspensions were prepared as described above for the auto-aggregation assay. Equal volumes (500 μL) of *P. pentosaceus* 1101 and *L. monocytogenes* 1639 were mixed and incubated at room temperature. Pure bacterial suspensions (1 mL each) were incubated under similar conditions as controls to check self-flocculation. The A_600nm_ of the mixture was determined at the indicated times (0, 2, 4, 6, 20, and 24 h), and the percentage of co-aggregation was calculated according to the equation of Malik [34]: C [%] = [(A_pat_ + A_probio_) − (A_mix_)]/[(A_pat_ + A_probio_)] × 100 (5)
where A_pat_ and A_probio_ represent the absorbance of the independent bacterial suspensions at 0 h, and A_mix_ corresponds to the absorbance of the bacterial mixture at the different times tested. 

### 2.5. Proteomic Analysis of P. pentosaceus 1101 Exposed under Control and Gastrointestinal Conditions

#### 2.5.1. Strain Exposure under Control and Gastrointestinal Conditions

*P. pentosaceus* 1101 was cultivated in MRS up to the late logarithmic growth phase (37 °C, 15 h). Cells were recovered by centrifugation at 6271× *g* for 10 min at 4 °C and then twice washed with 1X PBS cold buffer (pH 7.2) prior to being exposed to the treatments. The gastrointestinal-conditions treatment involved the resuspension of the cells into MRS medium (pH 3.0, with 0.5% (*w*/*v*) bile salts); they were then incubated for 1 h at 37 °C. The control treatment involved the cell resuspension in MRS (pH 7.0), and this was performed under the same conditions. After incubation, cell suspensions were quickly immersed in an ice-water bath (<4 °C) to ensure the complete cooling of the samples [35]. Finally, cells for both treatments were collected by centrifugation at 6271× *g* for 10 min at 4 °C, subjected to two consecutive washes with cold PBS buffer (pH 7.2), and stored at −70 °C for electrophoretic or proteomic analysis.

#### 2.5.2. SDS–PAGE Electrophoretic Analysis and Lytic Activity by Zymograms

##### Protein Extraction

To extract all proteins, recovered cells were thawed at 4 °C and resuspended in lysis buffer (50 M Tris-HCl, 6 μL phenylmethylsulphonyl fluoride, and a protease inhibitor). The suspensions were then placed in vials with 2 mm glass beads and pre-incubated at 0 °C. The cells were broken by using a disruptor (Mini-Beadbeater 16, Biospec Products, Bartlesville, USA), applying 25 successive cycles of 30 s at 4 °C. At the end of the mechanical rupture, the suspensions were sedimented at 12,096× *g* for 20 min at 4 °C to separate the supernatant from the glass beads. The soluble protein fraction was recovered in the supernatant, and proteins were precipitated at −20 °C for 6 h with 20% trichloroacetic acid (TCA) and centrifuged at 12,096× *g* for 15 min at 4 °C. The supernatant was discarded, and the protein pellet was washed twice with cold acetone, centrifuged at 6271× *g*, and finally washed with 80% acetone [36]. The protein pellet was resuspended in buffer (5 M Tris-HCl, 5 mM DTT, 6 μL of a 6 M urea, pH 8.0) to solubilize and rehydrate the proteins for 30 min, while stirring, at 37 °C and then sonicated (Ultrasonic Bath, Bransonic^®^ CPXH2800, state or city, USA) for 30 min at low frequency (40 kHz). The protein concentration was quantified by using the Bradford method, and the soluble protein fraction was stored at −70 °C prior to the electrophoretic analysis.

##### Electrophoretic Analysis

SDS–PAGE analysis of the extracted proteins was performed according to the technique described by Laemmli [37]. Protein samples were mixed with sample loading buffer 5X (100 mM Tris-HCl pH 6.8, 1% [*w*/*v*] SDS, 24% [*v*/*v*] glycerol, and 0.02% Coomassie blue). Extracts containing 25 μg of protein from each treatment were deposited on a 12% polyacrylamide gel. Meanwhile, the zymographic assay was performed in 12% T gel (acrylamide/bisacrylamide) with 0.2% freeze-dried cells of *Micrococcus lysodeikticus* ATCC 4698 as a substrate, with a crosslinking of 2.6% (C). The concentration gels were prepared at 2.6% C and 4% T. A molecular mass marker with a range of 2–250 kDa (Precision Plus Protein Dual, Xtra Standard, Bio-Rad, Hercules, CA, USA) was used. The protein profiles obtained in the SDS–PAGE and zymogram gels were digitized by using a Gel Documentation System (GelDoc™ XR+, Bio-Rad, Hercules, CA, USA) and analyzed by using the Image-Lab (v6.1, Bio-Rad, Hercules, CA, USA). The apparent molecular weight (MW) of each protein band obtained from electrophoresis and zymogram was calculated by using the Gel Doc photo documenter software ™ XR+ Imaging System (Bio-Rad, Hercules, CA, USA), and the apparent weights of the bands were interpolated by using the semi-log regression method that manages the software, using Bio-Rad Dual Xtra as a standard.

#### 2.5.3. Protein Identification Using Liquid Chromatography Coupled to Tandem Mass Spectrometry (LC–MS/MS)

##### Extraction of Intracellular Protein Extracts from *P. pentosaceus* 1101

The biomass from the treatments (100 mg) was subjected to two consecutive washes with cold PBS buffer (pH 7.2) and macerated with liquid nitrogen; the resulting powder was resuspended in 1 mL of TRIS Reagent Solution (Invitrogen, Thermo Fisher Scientific, Waltham, MA, USA). The protein extracts were homogenized and further clarified by centrifugation at 4 °C for 15 min at 12,096× *g*. Proteins were precipitated with a volume of 1.5 mL isopropanol (100%); the final precipitate was washed twice with acetone (100%), followed by washing with 80% acetone solution; and the protein pellet was resuspended in buffer (50 mM Tris-HCl, 5 mM DTT, 6 M Urea, pH 7.4) and lyophilized.

##### LC–MS/MS Analysis

Prior to LC–MS/MS, protein extracts were resolubilized in 10 µL of a 50 mM Tris-HCl buffer, pH 8, with 6 M urea. Proteins were reduced with 45 mM DTT and 100 mM ammonium bicarbonate for 30 min at 37 °C. Subsequently, iodoacetamide was added for protein alkylation (100 mM iodoacetamide and 100 mM ammonium bicarbonate for 20 min at 24 °C in dark). Proteins were digested with trypsin solution (5 ng/µL of trypsin sequencing grade from Promega, 50 mM ammonium bicarbonate). Protein digestion was performed at 37 °C for 18 h. The protein digests were desalted by using MCX (Waters Oasis MCX 96-well Elution Plate) (MA, USA). Peptides were loaded into a 75 μm inner diameter × 150 mm Self-Pack C18 column installed in the Easy-nLC II system (from Proxeon Biosystems, now Thermo Scientific). The solvents used for chromatography were 0.2% formic acid (Solvent A) and 90% acetonitrile/0.2% formic acid (Solvent B). Solvent B first increased from 1 to 37% in 85 min and then from 40 to 85% B in 15 min. Peptides were eluted with a two-slope gradient, at a flowrate of 250 nL/min. The HPLC system was coupled to an Orbitrap Fusion mass spectrometer (Thermo Scientific) through a Nanospray Flex Ion Source. Full-scan MS survey spectra (*m*/*z* 360–1560) in profile mode were acquired in Orbitrap with a resolution of 120,000 and a target value of 3e5. Peptide ions were fragmented in the HCD collision cell and analyzed in the linear ion trap with a target value of 2e4 and a normalized collision energy of 28.

##### Database Searching and Protein Identification Criteria

All MS/MS samples were analyzed by using Mascot (Matrix Science, London, UK; v2.6.2). Mascot was set up to search the Uni-prot_Pediococcus_Pentosaceus_strain_ATCC_25745 database. Mascot was searched with a fragment ion mass tolerance of 0.60 Da and a parent ion tolerance of 10.0 PPM. Scaffold (version Scaffold_5.1.2, Proteome Software Inc., Portland, OR, USA) was used to validate MS/MS-based peptide and protein identifications. Peptide identifications were accepted if they could be established at greater than 90.0% probability by the Peptide Prophet algorithm [38]. Protein identifications were accepted if they could be established at greater than 90.0% probability and contained at least 1 identified peptide [39]. A false discovery rate (FDR) of <1.0% was established based on a decoy database.

##### Bioinformatic Analysis

Total spectra were used for label-free protein quantification [40]. Gene ontology (GO) enrichment analysis was performed by using DAVID [41] and visualized by using R studio (version 2022.7.1.554, MA, EUA).

### 2.6. Statistical Analysis

All determinations were performed in triplicate, and the results were expressed as the mean ± the standard deviation. Significant differences were calculated through multiple comparisons of the Duncan’s test or two-sample *t*-test (*p* < 0.05); it was ensured that the obtained data would meet the assumption of homogeneity of variances. Statistical analysis was performed by using NCSS 2007 software (version 1) [26].

## 3. Results and Discussion

### 3.1. Identification of the Strain

The identity of the *P. pentosaceus* 1101 strain was determined by using two different techniques. The BLAST sequence similarity was obtained with 99% identity and 100% coverage with the *P. pentosaceus* DSM20336 16S rRNA gene, using the partial sequence with accession number NR_042058. The strain’s identity was corroborated by using MALDI Biotyper Systems with a high-confidence identification (2.27) and consistency, and the 16S rRNA sequence obtained was deposited in GenBank (NCBI), with the accession number *P. pentosaceus* 1101: MZ265376.

### 3.2. Genome Sequencing and Assembly

The final genome assembly compromised seven contigs and an overall GC content of 37.05%, obtaining an N50 value of 293,806 and genome size of 1.76 Mbp. The *P. pentosaceus* 1101 genome was annotated by using the Prokka pipeline, which identified 1754 coding sequences (CDS), 55 rRNAs, and 33 tRNAs, of which 576 corresponded to proteins identified through the proteomic analysis under GC and control treatments. The whole genome of *P. pentosaceus* 1101 was deposited in the DDBJ/ENA/GenBank, under the accession number JAOAND000000000, Bioproject PRJNA876950, and Biosample SAMN30676162. The results obtained from the genome assembly are consistent with reports that the genome size of the type of strain is 1.81 Mbp and it has a GC content of 38.1% [13].

### 3.3. Evaluation of Probiotic Properties

#### 3.3.1. Antagonistic Activity

Table 1 summarizes the results of the antagonistic activity of the *P. pentosaceus* 1101 strain. This strain showed activity against all tested microorganisms, both Gram-negative (*E. coli* and *S. enterica* subsp. *enterica* serovar Typhimurium) and Gram-positive (*L. monocytogenes*, *S. aureus*, and *E. faecalis).* This antagonistic activity is of particular interest in relation to the ability to compete with pathogenic and spoilage microorganisms and their ability to produce antibacterial effects in situ, although this will depend on factors such as pH and temperature, as well as on the presence of competing organisms or additive effects [42].

#### 3.3.2. Growth, Proteolytic and Inhibitory Activity at Different pH Values

Figure 1a shows the effect of the pH on the growth rate (μ) and maximum optical density (O.D.max) for *P. pentosaceus* 1101. The influence of the pH on growth was demonstrated by the increased growth rate as the pH value increased from 2 to 7.0 (0.639 ± 0.014). In this study, *P. pentosaceus* 1101 showed the highest proteolytic activity at a low pH (5.5) (Figure 1b). The inhibitory activity was favored under acidic conditions (pH 4.5–5.0), and the inhibitory activity for *L. innocua* was second best, at pH 7.0 (Figure 1c). These results provide the first documented approach to optimize the physicochemical growth conditions for improving the inhibitory activity of strain 1101.

*P. pentosaceus* 1101 grows in a pH range of 2–7; this fact could be related to the strain’s presence in fermented meats, as during the fermentation and drying of sausages, an increasingly acidic pH profile (5.68–4.96) stimulates the release of actin-derived peptides and favors the growth of LAB [43]. A similar condition was observed for the studied strain 1101, which has high proteolytic activity at pH 5.5 that decreases with the increase of pH, suggesting that *P. pentosaceus* 1101 could produce active proteolytic enzymes under acidic conditions. The pH affects the transport of several growth factors across the LAB cell membrane; therefore, if the bacterium is not at its optimal growth pH, its proteolytic activity will be affected directly or indirectly [44]. In this study, proteolytic activity was favored at an acidic pH of 5.5, as well as pH = 7; other results demonstrated a similar behavior with moderate anti-listerial activity of *P. acidilactici* at pH 3.5 and 8.5, at a temperature of 30 °C [45]. The antagonistic activity exhibited by *P. pentosaceus* 1101 in this study may be related to the acidic environment caused by the production of organic acids that inhibit the development of altering microorganisms at low pH values [46]. However, several studies have reported that *P. pentosaceus* produces antimicrobial compounds, such as peptides, bacteriocins, and PGH, so the antimicrobial activity may be due to the combined effect of these metabolites [45,46,47]. The efficacy of these antimicrobial peptides may depend on several factors, including the pH, the presence of endogenous proteolytic enzymes, the membrane potential, and the lipid composition, that ultimately induce changes in the antimicrobial activity and their specificity. For instance, bacteriocins are stable in a wide range of pH values, with high activity at neutral and basic pH, but they are inactivated by endogenous proteases [47]. Thus, the observed higher antimicrobial activity at pH 5.5 and 7 for *P. pentosaceus* 1101 coincides with a decrement of the proteolytic activity.

#### 3.3.3. *P. pentosaceus* 1101 Survival after Exposure to Low pH and Bile Salts

Survival under the gastrointestinal tract conditions of low pH and bile salts is essential for microbial strains to be considered probiotics [48]. Before reaching the distal part of the gastrointestinal tract and exercising its probiotic effects, these bacteria must survive during the transition from the stomach to the upper part of the gastrointestinal tract. The pH values in the human stomach range from 1.5 during fasting to 4.5, and food intake can take up to 3 h [49]. Therefore, it is necessary to evaluate survival and growth at a low pH of 2.0 and 3.0 and 0.5% bile salts as preliminary assessments of the tolerance of *P. pentosaceus* 1101. Table 2 presents the kinetic parameters and shows that the studied strain is acid-resistant, and the addition of bile salts favors its growth rate.

Resistance to acidic pH and tolerance to 0.5% bile salts are conditions that affect bacterial survival during passage through the gastrointestinal tract, as reported by Martoni, using an in vitro model [50]. Strain 1101 survived exposure for 1 h at pH 2.0 and pH 3.0 and tolerated the conditions of 0.5% bile salts for 1 h. At pH 2.0, there was a decrease of 1.201 log_10_ CFU/mL; both at pH 3.0 and in the medium supplemented with 0.5% bile salts, there was a much smaller decrease to a logarithmic unit, starting from an initial count of 9.0 log_10_ CFU/mL. The data show that the average acidic logarithmic survival rate of *P*. *pentosaceus* 1101 at pH 2.0 was 86.8%, and it was 93.9% at pH 3.0, both in the medium with no bile salts. However, when the medium contained 0.5% bile salts, the logarithmic survival rate was 99.1% (Table 2).

Strain 1101 showed tolerance to acidity and bile salts, and its resistance to bile salts is related to bile salt hydrolase (HSB) activity, which can hydrolyze the combined biliary salt and thus reduce its toxicity and side effects [51]. The HSB is encoded by all sequenced intestinal lactobacilli and facilitates adaptation to the human gut [52]. In addition, the *P. pentosaceus* 1101 strain showed resistance to acidity after incubation at pH 2 and 3 for 3 h, similarly to that reported for the commercial probiotic strains *Lactobacillus acidophilus* LA-1 and *L. rhamnosus* GG, which were used as references for the estimation of gastrointestinal transit tolerance [53,54]. The results of logarithmic survival rate in acidity and bile salts that were obtained in this study are comparable with other previously documented for the reference strain *L. acidophilus* NCFM^®^, with an 81% of logarithmic survival rate reported at pH 2 and 89.2% in 0.5% of bile salts [55]. In addition, other reports with the strains *L. acidophilus* ADH [53] and *Bifidobacterium* [51] also resist gastrointestinal conditions with a decrease in 1 log_10_ CFU/mL, like the obtained data for *P. pentosaceus* 1101 show under similar conditions.

#### 3.3.4. Antibiotic Resistance

Antibiotic resistance evaluated with *P*. *pentosaceus* 1101 is shown in Table 3. The studied strain was inhibited to a sensitive extent by amoxicillin, clavulanic acid, penicillin, cefazolin, cefuroxime, and chloramphenicol, but the strain shows itself to be resistant to cephalexin, piperacillin, azithromycin, erythromycin, tetracycline, ciprofloxacin, ofloxacin, and cotrimoxazole. Resistance to erythromycin and tetracycline has been reported particularly in LAB [56]. It is noteworthy that despite the possible resistance of *P. pentosaceus* 1101 to most of the antibiotics tested, it was sensitive to clinically important antibiotics, such as penicillin, amoxicillin, and chloramphenicol, and this would allow it to be considered relatively “safe”. These findings are consistent with those reported by Savedbowoen [31], who found that most tested strains of *P. pentosaceus* were sensitive to penicillin, chloramphenicol, and erythromycin, and those reported for a strain of *P. pentosaceus* had potential as a vaginal probiotic [57]. Current knowledge on antibiotic resistance in LAB remains limited, and the controversy regarding the safety of the use of potential probiotic strains with some antibiotic resistance has been widely discussed, with some authors arguing that probiotic bacteria possessing antibiotic resistance profiles could be an advantage in clinical applications by facilitating their co-administration with antibiotic treatment [58]. In addition, microbial isolates intended to be used as probiotics should also display some antibiotic resistance in order to survive in the gastrointestinal tract of the host and to exert beneficial activities on it [6]. However, it has been shown that probiotic strains can reduce the number of antibiotic-resistance genes in colonization-permissive and antibiotic-naïve individuals, which may vary depending upon age, food habits, and health conditions [59,60]. After a course of antibiotics treatment, these probiotic strains may exacerbate the antibiotic-mediated resistome expansion in the gastrointestinal tract but do not contribute to the increase in antibiotic resistance genes from their own repertoire by horizontal transfer to commensals and pathogens. The studied strain shows resistance to the most common antibiotics, and there is a very low possibility for transfer of these genes by conjugation [59]. Moreover, in this study, we did not identify any vancomycin resistance genes in *P. pentosaceus* 1101.

#### 3.3.5. Cell Adhesion Properties

In the present study, the affinity for the organic solvents hexane and chloroform was assessed to determine the cell-surface properties. The affinity to hexane (non-polar solvent) demonstrates the characteristic hydrophobic surface of the bacteria, and the affinity to chloroform (polar acid solvent) describes the electron donor property of the bacterial cell surface, which is attributed to carboxylic groups and the Lewis acid–base interactions [61]. The studied strain showed (Table 4) significantly higher hydrophobicity (91.5 ± 0.3%) than the control *L. acidophilus* NCFM^®^ (51.82 ± 0.3%); both strains have affinity for chloroform, with *L. acidophilus* NCFM^®^ having significantly higher affinity (94.06 ± 0.5%), indicating its electron-donating capacity. These results show that *P. pentosaceus* has a profile of adherence to hydrophobic surfaces. Some reports have suggested that bacterial cells with high hydrophobicity tend to form strong interactions with mucosal cells or adhere strongly to epithelial cells [5,6].

The auto-aggregation capacity of the studied strain reached 87%, and its co-aggregation with the pathogen *L. monocytogenes* 1639 exceeded 74% after 24 h of incubation (Table 5). The antagonistic properties of probiotic strains are also related to the formation of co-aggregates with pathogens that hinder their adhesion to the intestinal epithelium and form a microenvironment that helps probiotics to eliminate pathogenic microorganisms. [6].

### 3.4. Proteomic Analysis of P. pentosaceus 1101 Exposed under Control and Gastrointestinal Conditions

#### 3.4.1. SDS–PAGE Analysis and Zymographic Antimicrobial Activity Determinations

To establish the variation in the production of protein metabolites with antimicrobial activity associated with gastrointestinal culturing conditions, it was necessary to determine the profile of soluble proteins present under normal conditions for *P. pentosaceus* 1101 to compare with the protein profile obtained under gastrointestinal conditions.

Figure 2a shows the protein profile obtained from the soluble protein fraction for control and treatment. It has been observed for the genus *Pediococcus* spp. that the expression of antimicrobial proteins is variable according to pH conditions [62]. Bands were detected from 10 to 250 kDa; the protein profile was affected by changes in pH and the presence of bile salts, as some bands disappeared from 10 to 25 kDa and some of the intensity decreased from 75 to 250 kDa compared to the control. A band with lytic activity of the intracellular fraction was detected in the gastrointestinal-conditions treatment with a molecular weight of 29 KDa (Figure 2b).

#### 3.4.2. Protein Identification Using Liquid Chromatography Coupled to Tandem Mass Spectrometry (LC–MS/MS)

A total of 576 proteins were identified, and a comparison between the proteins identified in the control treatment and gastrointestinal conditions treatment groups through proteomic analysis revealed that 493 proteins were expressed in both groups, 63 proteins were expressed only in the control treatment group, and 20 only in the gastrointestinal condition’s treatment group (Figure 3a). Based on sequence homology, proteins were classified into functional groups. A circular bar plot shows the top 10 functional groups in each of the 3 categories of gene ontology classification (Figure 3b), namely biological process, cellular component, and molecular function, representing the assignment. In total, 19% of the identified proteins were associated with the biological process of translation, followed by 4% for the carbohydrate metabolic process. In addition, 46% of the proteins were associated with the cellular component of the cytoplasm, and 16% were for the integral component of the membrane and ribosome. In terms of molecular function, most of the proteins were associated with ATP binding and ion binding.

Several studies report on the adaptation of LAB to different environmental stresses that could be due to underlying changes in gene transcription and protein expression. [54]. The adaptation the mechanism of LAB under stressful conditions involves the upregulation of different stress-responsive proteins [5]. A quantitative analysis using a *t*-test between the gastrointestinal conditions treatment and control groups yielded 100 proteins that showed significant differential abundance; 73 of these proteins were downregulated, and 27 were upregulated (Figure 4). These results suggest that exposure to 0.5% bile salts and acidity in the treatment influence the protein profiles of *P. pentosaceus* 1101.

Table 6 shows the impact of exposure to 0.5% bile salts and acidity on the proteomic profile, summarizing the 18 main upregulated proteins and those found only in the gastrointestinal-conditions treatment group involved in tolerance processes to gastrointestinal conditions, belonging to the following functional categories: two-component regulatory system, peptidoglycan catabolic and biosynthetic process, proteolysis, DNA repair, response to stress, cell adhesion, and penicillin binding.

Two of these proteins involved in tolerance processes to gastrointestinal conditions, the signal transduction histidine-protein kinase ArlS and the serine/threonine protein kinase, are involved in signal transduction; signaling through protein kinases is evolutionarily known [63]. The signal transduction histidine-protein kinase ArlS has been correlated with the two-component signal transduction system (TCS), which is the predominant way in which bacteria sense and respond in order to adapt to environmental changes, consisting of a histidine receptor kinase (ArlS) and a response regulator (ArlR) [64]; this ArlS-ArlR locus has been shown to affect PGH expression, proteolytic activity, and cell-adhesion properties, so this same TCS system may be involved in both antimicrobial activity and cell physiology [65]. Serine/threonine protein kinase is a characteristic feature of many Gram-positive bacteria; these proteins are involved in vital processes, such as cell division and DNA replication, and interact with epithelial cells in the human gut, induce immune response, and modulate antibiotic resistance [63].

The proteomic analysis also identified PGH *N*-acetylmuramoyl-L-alanine amidase (32 kDa); the *N*-acetylmuramoyl-L-alanine amidase enzyme domain cleaves the amide bond between *N*-acetylmuramoyl and L-amino acid residues of glycopeptides in bacterial cell walls and has an N-terminal Src Homology 3 (SH3) protein interaction domain. Its structure shows that two conserved glutamic acids are the main catalytic residues, and zinc acts as a cofactor [66]. In general, *N*-acetylmuramoyl-L-alanine amidases are members of the autolytic bacterian system and carry a signal peptide at their N-terminus that enables their transport across the cytoplasmic membrane, and the catalytic module of the amidase is fused to another functional module at the N- or C-terminus, which is responsible for the high-affinity binding of the protein to the cell wall [67]. The influence of the pH and the presence of bile salts on the PGH profile in the 1101 strain were observed since the band identified as *N*-acetylmuramoyl-L-alanine amidase (32 kDa) was detected by zymography in treatment with a molecular weight of 29 kDa. The difference in molecular weights observed between the zymogram band (29 kDa) and that obtained from the proteomic analysis (32 kDa) may be due to the differential electrophoretic mobility in the PAGE containing bacterial cells (*M. lysodeikticus*), which are lysed and traditional SDS-PAGE. Furthermore, constituent polypeptides exhibit PGH activity; thus, the observed extracellular PGH may be released from bacterial cell walls [68]. The profile of PGH visualized in this study did not allow for the determination of the exact number of these enzymes produced by *P. pentosaceus* 1101 because only enzymes that can be renatured and that retain enzymatic activity are detectible [69,70]. These results suggest that the antimicrobial activity of *Pediococcus* spp. may be due to PGH production, as lytic activities of high and low molecular weight have been reported [11,71,72]. Finally, the presence of a lytic band in acidity and bile salt treatment obtained from the cytosolic soluble protein fraction in zymogram and identified through the proteomic analysis as *N*-acetylmuramoyl-L-alanine amidase was confirmed to have a role in managing defenses under gastrointestinal conditions.

The harmful effects of bile include detergent action, oxidative, low pH, and osmotic stress [73]. In our treatment, we identified the upregulated chaperone protein DnaJ, which actively participates in the response to hyperosmotic and thermal shock by preventing the aggregation of stress-denatured proteins and disaggregating unfolded proteins that initially bind to DnaJ [74]. An ATPase component of the ABC transporter has also been identified; this protein is known to be involved in the response of BAL to osmotic stress. It transports non-noxious compounds, allowing for cellular detoxification, one of the deleterious effects of bile [73].

Consequently, when *P. pentosaceus* 1101 was exposed to gastrointestinal conditions treatment, 73 proteins were downregulated (Table 7). Overall, these data show that the exposure of the tested strain to gastrointestinal conditions treatment results in significant global alterations in carbohydrate metabolism and transporter activity.

ABC-type multidrug transport systems are proton/sodio motive force transporters that use the energy of ATP hydrolysis to extrude various substrates, including drugs and antibiotics, across the cell membrane [75]. *P. pentosaceus* 1101 was found to have some antibiotic resistance, as proteins related to antibiotic transport and binding were identified, such as the ABC-type multidrug transport system (downregulated) and the cell elongation-specific peptidoglycan D,D-transpeptidase (penicillin binding), which was found to be upregulated; however, the deregulation of the two proteins involved in antimicrobial and antibiotic transport in treatment would indicate that this type of transport is affected under stress gastrointestinal conditions. In the proteomic analysis, we did not identify factors associated with the pathogenicity of the strain with the searching tools, using keywords such as “toxin, phage proteins, transposase, sortase, collagenase, hyaluronidase S, BspA, PrfA, PlcH, AvrA, Cnf1, VanC1, VanC2, ErmA, ErmB, ErmC, GelE, Mef, Lnu, CylL, Ccf, FsrB” [16]. As far as we can say, under the tested conditions, *P. pentosaceus* 1101 did not express genes related to virulence.

We observed that the stress defense mechanisms described for the *P. pentosaceus* 1101 strain are grouped according to their direct involvement in cellular energy conservation, macromolecule defense, and cell envelope protection [62]. These proteins identified in the proteomic analysis involved in the response to acid stress and bile salts are also involved in the adaptive response to technological stress conditions. We found chaperones in the proteomic analysis, which also indicate the inherent thermotolerance capacity of the studied strain, as it grows at 43 °C (see Appendix A), which is important for the production of cooked food products, and different LAB species that have been isolated from cooked sausages show this capacity [76]. Through the proteomic analysis, we were able to identify a periplasmic protease in the stress treatment, and the proteolytic activity of the tested strain was favored at an acidic pH of 5.5, which corroborates the production of proteolytic enzymes active under acidic conditions. Many pathogenic and food-spoilage microorganisms are sensitive to high osmotic pressure compared to LAB. NaCl is often added to help LAB initiate the fermentation process. This osmotic stress decreases the positive turgor of bacterial cells as a result of dehydration; under such conditions, the cells produce or import small molecules, called osmolytes, to balance the difference between intracellular and extracellular osmolarities and allow rehydration through membrane-associated channels [62]. *P. pentosaceus* 1101 grew at a NaCl concentration of 1–6% and showed proteolytic and inhibitory activity at these concentrations (see Appendix A). In this study, we identified the protein ATPase component of the ABC transporter, which, as mentioned, is involved in response to osmotic stress. It is believed that NaCl concentrations of up to 7% can enhance the acid-producing activity of LAB and antimicrobial compounds [45,77].

## 4. Conclusions

It was probed that several proteins may be a key factor in the adaptation of *Pediococcus pentosaceus* 1101 in the gastrointestinal tract; this strain that was isolated from Spanish-type chorizo shows the ability to survive acidity and bile salts, and it possesses cell adhesion properties and some antibiotic sensitivity. Most of the proteins that were identified were found to be associated with biological translation processes, carbohydrate metabolism, and molecular functions of ATP and ion binding. The adaptation mechanism of *P. pentosaceus* 1101 under stress gastrointestinal conditions involved the differential regulation of proteins involved in the mechanisms by which the strain senses the effects of acid and bile salt and responds to environmental changes (histidine-protein kinase ArlS signal transduction), cell adhesion (serine/threonine protein kinase and surface adhesin), modulation of antibiotic resistance (multidrug and ABC-type antimicrobial transport system), antimicrobial activity (*N*-acetylmuramoyl-L-alanine amidase), osmotic stress (ATPase component of ABC transporter), and deaggregation of unfolded proteins (DnaJ chaperone protein). *P. pentosaceus* 1101 also showed significant antagonistic, inhibitory, and proteolytic activity, thus signifying that the strain could promote the hydrolysis of food proteins during the fermentation or ripening process and intervene in the development of sensory characteristics of the products, as well as microbiological stability and safety. Based on the results of the present study, *Pediococcus pentosaceus* 1101 could have potential for use as a probiotic, as it can survive the passage to the gut, or could be used as a bioprotective culture in fermented foods due its ability to compete with spoilage and pathogenic microorganisms. We obtained a series of results that provide information on how probiotics respond molecularly to stress during gastrointestinal transit and that could also be useful for the selection of strains involved in food processing.

## Figures and Tables

**Figure 1 foods-12-00046-f001:**
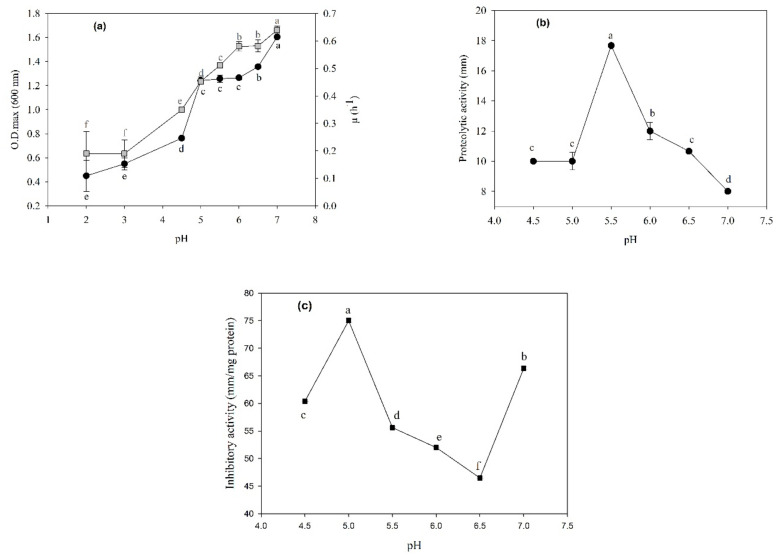
(**a**) Influence of pH (pH 2 and 3 for 9 h; pH 4.5–7 for 24 h) on the growth of *P*. *pentosaceus* 1101: O.D.max (●), μ (■). (**b**) Proteolytic activity curve. (**c**) Inhibitory activity with *Listeria innocua*. The determinations were performed in triplicate, and the values represented are the mean with standard deviation of three independent experiments performed with three different cultures. Different letters indicate significant differences (*p* < 0.05) according to Duncan’s multiple comparison of means.

**Figure 2 foods-12-00046-f002:**
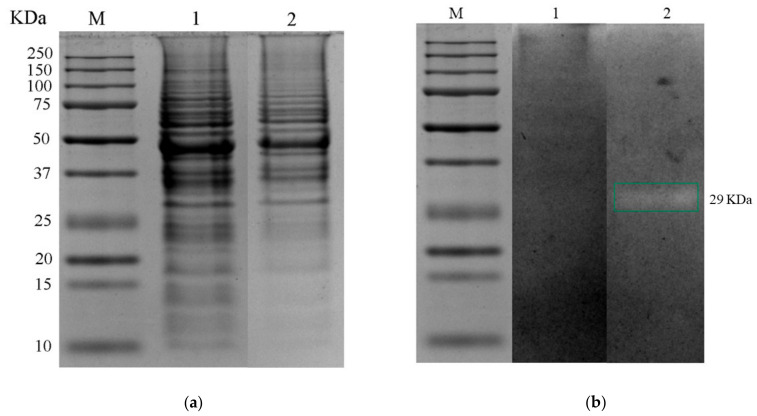
(**a**) SDS–PAGE 12% and (**b**) zymogram of soluble fraction of intracellular proteins obtained with the different treatments. M—molecular-weight marker Bio-Rad Dual Xtra; Lane 1—control treatment; Lane 2—gastrointestinal-conditions treatment.

**Figure 3 foods-12-00046-f003:**
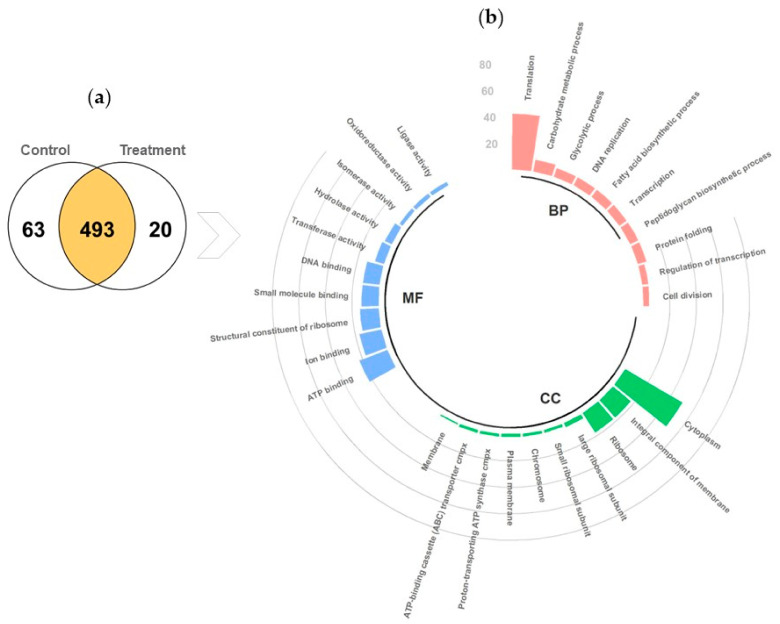
(**a**) Venn diagram and (**b**) circular bar plot: gene ontology classification for global proteins. The results are summarized in three main categories: biological process (**BP**), cellular component (**CC**), and molecular function (**MF**).

**Figure 4 foods-12-00046-f004:**
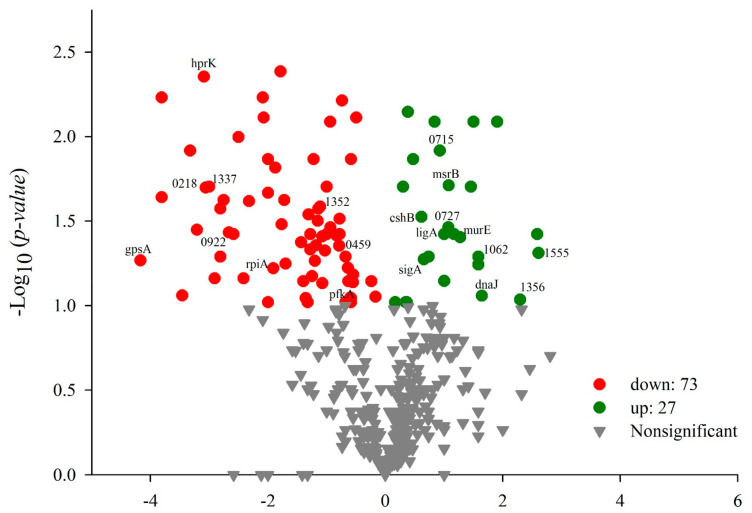
Volcano plot: change level of treatment and control proteins. Red dot, downregulated; green dot, upregulated; and gray dot, nonsignificant (*t*-test *p* < 0.1).

**Table 1 foods-12-00046-t001:** Antagonistic activity of *P*. *pentosaceus* 1101 against different test microorganisms.

Strain	Antagonistic Activity
*Escherichia coli* DH5α GenBank:NZ_JRYM00000000.1 ^a^	+
*Enterococcus faecalis* 1351 ^b^	+
*Listeria monocytogenes* 1639 ^b^	+
*Salmonella enterica* subsp. *enterica* serovar Typhimurium ATCC 14,028 ^a^	+
*Staphylococcus aureus* ATCC 6538 ^a^	+
*Weisella viridescens* GenBank: UAM-MG5: MT814884 ^b^	+

+ Indicates positive reaction by inhibition of the growth of the test strain. Strain source: ^a^ Department of Food and Biotechnology, Faculty of Chemistry, Universidad Nacional Autónoma de México; ^b^ Department of Biotechnology, Universidad Autónoma Metropolitana, Mexico.

**Table 2 foods-12-00046-t002:** Evaluation of tolerance to acidity and bile salts: survival rate after exposure to bile salts and pH 2 and 3 for 1 h.

Condition	O.D.max	µ (h^−1^)	Initial Count log_10_ CFU/mL	Final Count log_10_ CFU/mL	Logarithmic Survival Rate (%)
	*Resistance to bile salts*				
Control	1.30 ± 0.01	0.69 ± 0.01 ^b^			
0.5%	1.28 ± 0.01	1.08 ± 0.01 ^a^	9.448 ± 0.005	9.352 ± 0.002	99.1
	*Resistance to acidity*				
Control	1.26 ± 0.01 ^a^	0.72 ± 0.01 ^a^			
pH 2	0.45 ± 0.13 ^b^	0.19 ± 0.08 ^b^	9.115 ± 0.012	7.914 ± 0.003	86.8 ^b^
pH 3	0.55 ± 0.05 ^b^	0.19 ± 0.05 ^b^	9.115 ± 0.012	8.556 ± 0.024	93.9 ^a^

The values represented correspond to the mean values and standard deviations of three independent experiments. Student’s *t*-test two-sample comparison analysis of means was performed, and differences were considered statistically significant at *p* < 0.05. Different letters indicate a significant difference between values within the same column.

**Table 3 foods-12-00046-t003:** Antibiotic-resistance profile.

Antibiotic	Resistance Profile
*Cell-wall-biosynthesis inhibitors*	
Amoxicillin/Clavulanic acid	S
Amoxicillin	S
Penicillin	S
Cephalexin	R
Cefazolin	S
Cefuroxime	S
Piperacillin	R
*Protein-synthesis inhibitors*	
Azithromycin	R
Erythromycin	R
Chloramphenicol	S
Tetracycline	R
*DNA-synthesis inhibitors*	
Ciprofloxacin	R
Ofloxacin	R
*Metabolic products inhibitor*	
Cotrimoxazole	R

S, sensitive (≥21 mm); R, resistant to the tested antibiotic (≤15 mm) [31].

**Table 4 foods-12-00046-t004:** Bacterial hydrophobicity (%) of LAB.

LAB	Solvent	
	Hexane (%)	Chloroform (%)
*P. pentosaceus* 1101	91.5 ± 0.3 ^a^	19.6 ± 0.4 ^b^
*L. acidophilus* NCFM^®^	51.8 ± 0.3 ^b^	94.1 ± 0.5 ^a^

The presented values correspond to the mean values and standard deviations of three independent experiments. A two-sample *t*-test comparison analysis of means was performed, and differences were considered statistically significant at *p* < 0.05. Different letters indicate significant differences between values within the same column.

**Table 5 foods-12-00046-t005:** Auto-aggregation/co-aggregation capacity (%) of LAB.

LAB	2 h	4 h	6 h	20 h	24 h
*P. pentosaceus* 1101	10.7 ± 0.3 ^b^/51.2 ± 0.2 ^b^	14.7 ± 0.5 ^b^/56.0 ± 0.2 ^b^	21.4 ± 0.4 ^b^/60.1 ± 0.2 ^b^	62.9 ± 0.5 ^b^/64.4 ± 0.1 ^b^	86.7± 0.2 ^b^/74.3 ± 0.2 ^b^
*L. acidophilus* NCFM	14.3 ± 0.2 ^a^/56.3 ± 0.1 ^a^	25.0 ± 0.1 ^a^/58.7 ± 0.1 ^a^	32.2± 0.3 ^a^/71.9 ± 0.2 ^a^	84.8 ± 0.3 ^a^/80.1 ± 0.1 ^a^	96.4 ± 0.2 ^a^/85.2 ± 0.2 ^a^

The presented values correspond to the mean values and standard deviations of three independent experiments. A two-sample *t*-test comparison analysis of means was performed, and differences were considered statistically significant at *p* < 0.05. Different letters indicate significant differences between values within the same column.

**Table 6 foods-12-00046-t006:** List of the top 18 proteins involved in tolerance processes to gastrointestinal conditions in *P. pentosaceus*.

Protein	Gene	Molecular Weight (kDa)	Function
Signal transduction histidine-protein kinase ArlS ^a^	*0715*	57	Two-component regulatory system
Serine/threonine protein kinase ^b^	*0832*	57	Phosphorylation
*N*-acetylmuramoyl-L-alanine amidase ^b^	*1117*	32	Peptidoglycan catabolic process
Peptidoglycan transpeptidase, ErfK-YbiS-YhnG family ^a^ UDP-*N*-acetylmuramoyl-L-alanyl-D-glutamate--L-lysine ligase ^a^	*1555* *murE*	51 56	Peptidoglycan biosynthetic process
LysM-domain-containing protein	*1356*	23	Hydrolase activity
Periplasmic protease ^a^	*1062*	52	Proteolysis
ATPase component of ABC transporter with duplicated ATPase domains ^b^	*1071*	72	ATP binding
RNA polymerase sigma factor SigA ^a^	*sigA*	43	DNA binding
DNA ligase ^a^	*ligA*	75	DNA repair
Single-stranded-DNA-specific exonuclease RecJ ^b^	*1130*	87	
Transcription-repair-coupling factor, TRCF ^b^	*mfd*	132	
Chaperone protein DnaJ ^a^	*dnaJ*	40	Response to stress
Peptide methionine sulfoxide reductase MsrB ^a^	*msrB*	17	
DEAD-box ATP-dependent RNA helicase CshB ^a^	*cshB*	52	
Protein RadA ^b^	*radA*	50	
Surface adhesin ^b^	*0086*	34	Cell adhesion
Cell-elongation-specific peptidoglycan D,D-transpeptidase ^a^	*0727*	74	Penicillin binding

^a^ Upregulated proteins. ^b^ Proteins present only in the treatment.

**Table 7 foods-12-00046-t007:** List of the top 13 downregulated proteins and those found in the control treatment only.

Protein	Gene	Molecular Weight (kDa)	Function
Glycerol-3-phosphate dehydrogenase ^a^	*gpsA*	36	Carbohydrate metabolic process
HPr kinase/phosphorylase ^a^	*hprK*	35	
Fructose-bisphosphate aldolase ^a^	*1352*	31	
Glyceraldehyde-3-phosphate dehydrogenase ^a^	*0459*	37	
ATP-dependent 6-phosphofructokinase ^a^	*pfkA*	34	
Ribose-5-phosphate isomerase ^a^	*rpiA*	25	
L-lactate oxidase ^a^	*0922*	40	Oxidoreductase
Malate dehydrogenase (NAD) ^a^	*0218*	33	Carboxylic acid metabolic process
ABC-type multidrug transport system ^a^	*1337*	64	Transporter activity
Energy-coupling factor transporter ATP-binding protein EcfA ^b^	*1390*	31	
ABC-type antimicrobial peptide transport system ^b^	*1650*	71	Antibiotic resistance
Predicted esterase of the alpha-beta hydrolase superfamily ^b^	*0212*	33	Lipid catabolism
Aspartate racemase ^b^	*0662*	27	Nitrogen compound metabolism

^a^ Downregulated proteins, ^b^ Proteins present only in the control treatment.

## Data Availability

All data associated with this manuscript are provided herein. The16S rRNA gene sequence obtained from *P. pentosaceus* 1101 was deposited in GenBank (NCBI), with the accession number *P. pentosaceus* 1101: MZ265376. (https://www.ncbi.nlm.nih.gov/nuccore/2043666188, accessed on 16 December 2022). The whole genome of *P. pentosaceus* 1101 was deposited in DDBJ/ENA/GenBank, under the accession number JAOAND000000000, Bioproject PRJNA876950 and Biosample SAMN30676162. The mass spectrometry data for protein identification can be found in the Appendix A.

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
