# Peer review of "Probiotic Properties and Proteomic Analysis of Pediococcus pentosaceus 1101"

_foods, 2022, doi:10.3390/foods12010046_

Round 1

Reviewer 1 Report

This study intended to assess the potential probiotic properties of Pediococcus pentosaceus 1101 by exposing this strain to conditions similar to the gastrointestinal tract and to elucidate its potential probiotic properties from a proteomic perspective. Although the experiment was not so innovative, the overall experimental design was comprehensive and well.

Line 28-29, delete fermented products and starter culture in keywords.

Line 46-48, probiotics should not become resistant to antibiotics, please modify this sentence.

Line 67-68, what is BAL? The class and family names of bacteria need to be italicized.

Line 78-79, please introduce the reason why 1101 strain was selected as the evaluation object in this experiment, that is, what potential advantages it has compared with other strains is worth further study.

Line 144, is there any theoretical basis for why the activation temperature is deliberately set at 35 ℃instead of 37 ℃.

Line 177, please revise the format of “O.D. 600nm=0.3”.

Line 220, why did authors choose cell hydrophobicity, auto-aggregation and co-aggregation capacities to evaluate the cell adhesion of P. pentosaceus 1101 instead of directly using the Caco-2 cell-mimicking intestinal epithelial cell model? What are its advantages? Please discuss.

Line 281, please revise “6,271 ×-g

Line 393, 414, 486, please change “work” into “study”.

Line 395-396 (Figure 1c), why does the trend of inhibitory activity against L. innocua differ at pH 4.5-7.0, I mean, why does the trend increase, then decrease, then increase again? Please explain.

Line 399, the fill color of the symbol is wrong; how to define μ and how to calculate μ in this study?

Line 429, how we can see from Table 2 that the addition of bile salts is beneficial to the growth rate of 1101?

Line 439, it is well known that lactic acid bacteria of natural origin have difficulty surviving at pH = 2 and 0.5% bile salts. The tolerance of strain 1101 in this study to acid and bile salts was 86.8%-99.1%, which is very amazing. Please check the test results again and give an appropriate explanation. Why was a control strain not used in this study, such as L. acidophilus LA-1 and L. rhamnosus GG?

Line 510, 511, please merge Table 5 and Table 5 into one table.

Author Response

The authors are grateful to the reviewers for their  comments to the manuscript, we present a new version of the manuscript that includes all the received  recommendations as follows:

Line 28-29, delete fermented products and starter culture in keywords.

Answer: were deleted from the text. Line 28-29.

Line 46-48, probiotics should not become resistant to antibiotics, please modify this sentence.

Answer: the sentence was modified . Line 45

Line 67-68, what is BAL? The class and family names of bacteria need to be italicized.

Answer: corrections have been made and the correct text is "LAB". Lines 70-72.

Line 78-79, please introduce the reason why 1101 strain was selected as the evaluation object in this experiment, that is, what potential advantages it has compared with other strains is worth further study.

Answer: The strain of P. pentosaceus 1101 was identified and selected for evaluation due to its wide inhibition spectrum, high resistance to acidic conditions and prevalence throughout the maturation process of a fermented meat product (Spanish type chorizo) from which it was previously isolated by our group [19].Lines 90-93

Line 144, is there any theoretical basis for why the activation temperature is deliberately set at 35 ℃instead of 37 ℃.

Answer:  the paragraph was corrected as the temperature is 37 °C. Line 172.

Line 177, please revise the format of “O.D. 600nm=0.3”.

Answer: was corrected by A600nm of 0.3. Line 225.

Line 220, why did authors choose cell hydrophobicity, auto-aggregation and co-aggregation capacities to evaluate the cell adhesion of P. pentosaceus 1101 instead of directly using the Caco-2 cell-mimicking intestinal epithelial cell model? What are its advantages? Please discuss.

Answer: Indeed, the best test is with the Caco-2 cell line, however, in our laboratory we do not have the conditions to work with cell lines. AS reported by other authors the cell surface hydrophobicity, auto-aggregation and co-aggregation capacities were employed for a preliminary screening of potential probiotic bacteria [33]. Line 278-282.

Line 281, please revise “6,271 ×-g”

Answer: was corrected by "6.271 × g". Line 340.

Line 393, 414, 486, please change “work” into “study”.

Answer: was changed in the text. Lines 486, 530 y 533.

Line 395-396 (Figure 1c), why does the trend of inhibitory activity against L. innocua differ at pH 4.5-7.0, I mean, why does the trend increase, then decrease, then increase again? Please explain.

Answer: The efficacy of these antimicrobial peptides may depend on several factors including the pH, the presence of endogenous proteolytic enzymes, the membrane potential, and the lipid composition, that ultimately induce changes in the antimicrobial activity and their specificity. For instance, bacteriocins are stable in a wide range of pH with high activity at neutral and basic pH, but they are inactivated by endogenous proteases [48].Thus the observed higher antimicrobial activity at pH 5.5 and 7 for P. pentosaceus 1101 coincides with a decrement its proteolytic activity. Lines 538-544

Line 399, the fill color of the symbol is wrong; how to define μ and how to calculate μ in this study?

Answer: the fill colour of the symbol was corrected (line 495). μ, is the growth rate. The kinetic parameters of growth rate (μ), and O.D.max were obtained by the following equation (line 190):

Where O.D.0 refers to the value of O.D. at time t=0. The minimum values of the square error were found as a function of the parameters O.D.0, O.D.max and μ. (lines 192-193.)

Also changed Fig. 1a, corrected Y-axis legend "O.D.max"( line 495.

Line 429, how we can see from Table 2 that the addition of bile salts is beneficial to the growth rate of 1101?

Answer: The results of the kinetic parameters obtained compared to the control are shown in table 2. Line 486.

Line 439, it is well known that lactic acid bacteria of natural origin have difficulty surviving at pH = 2 and 0.5% bile salts. The tolerance of strain 1101 in this study to acid and bile salts was 86.8%-99.1%, which is very amazing. Please check the test results again and give an appropriate explanation. Why was a control strain not used in this study, such as L. acidophilus LA-1 and L. rhamnosus GG?

Answer: We have corroborated all the data, it should be clarified that these values obtained of 86.8%-99.1% are calculated as a logarithmic survival rate, that is, the results obtained in CFU/ mL were converted to log10 to then calculate the %, this was done in order to compare with previous reports cited in the bibliography, with the following equation:

    (line 249)

Results of logarithmic survival rate in acidity and bile salts that were obtained in this study are comparable with other previously documented for the reference strain L. acidophilus NCFM®, with  an 81% of logarithmic survival rate reported at pH 2 and 89.2% in 0.5% of bile salts [56]. In addition, other reports with the strains L. acidopbilus ADH [54] and Bifidobacterium [52] also resist gastroin-testinal conditions with a decrease  in 1 log10 CFU / mL, like the obtained data for P. pentosaceus 1101 under similar conditions. Lines 662-667.

Line 510, 511, please merge Table 5 and Table 5 into one table

Answer: the tables were merged. Line 567.

Reviewer 2 Report

The article describes the identification of Pediococcus pentosaceus 1101 using 16S rRNA and MALDI-Biotyper. The strain was characterized for probiotic properties, including growth kinetics, proteolytic and inhibitory activities, survival at low pH and bile salt, antagonistic activity, cell-adhesion properties, and antibiotic resistance. This was followed by genomic and proteomic analysis, including identifying proteins obtained under control and gastrointestinal conditions. The results are interesting and related to detecting lytic enzyme and proteins responsible for adaptation in GIT. The strain has a good survival rate in GIT conditions. The research is relevant and interesting. The paper is well-written, clear, and easy to understand. I think the subject is overall interesting.

There is no data regarding the safety evaluation of the strain for probiotic characterization. The Introduction section needs improvement to explain the research gap and previously published work on the proteomics of P. pentosaceus.

Author Response

The authors are grateful to the reviewers for their  comments to the manuscript, we present a new version of the manuscript that includes all the received  recommendations.

Q. There is no data regarding the safety evaluation of the strain for probiotic characterization. The Introduction section needs improvement to explain the research gap and previously published work on the proteomics of P. pentosaceus

Answer: We did not identify in the proteomic analysis factors associated with the pathogenicity of the strain, with the searching tools using keywords such as "toxin, phage proteins, transposase, sortase, collagenase, hyaluronidase S, BspA, PrfA, PlcH , AvrA, Cnf1, VanC1, VanC2, ErmA, ErmB, ErmC, GelE, Mef, Lnu, CylL, Ccf, FsrB" [16]. As far as we can say that under the tested conditions P. pentosaceus 1101 did not expressed genes related to virulence. Lines 946-951.

The introduction was improved by including previously published work on the proteomic analysis of Pediococcus pentosaceus:

Proteomic methos based on LC-MS/MS offer advantages in speed and reliability compared to other traditional methods and allows the possibility to identify and characterize the global bacteria proteome, including the search of factors associated to pathogenicity as well as to antibiotic sensibility [16]. Some reports related to the proteomic analysis of P. pentosaceus have shown that this bacteria is capable to modulate its proteome when exposed to different stressful environments such as heat, cold, acid, bile and oxidative stress; in addition some proteins have been identified and could be used as markers to assess the stress tolerance, that may help to understand the stress tolerance mechanisms, thus providing new perspectives for the production of enhanced probiotics [17,18].Lines 81-88.

Reviewer 3 Report

The present manuscript provides valuable knowledge on how probiotics respond molecularly to stress during gastrointestinal transit.

I have the following comments:

Line 90: Provide the accession number of the isolates.

Lines 92-94: how did the authors confirm the efficiency of the preservation and reactivation methods? reference is required.

Line 107: PCR conditions should be detailed

Lines 195-199: according to the authors, a method by Khan [26] was employed with some modification. Nothing is mentioned about this adjustment in lines 195-199.

Line 208: Clarify how the authors obtained swine bile.

Lines 351-354: Does data meet the assumption of homogeneity of variances? Clarify if the authors run a homogeneity test.

Lines 395-396: the best inhibitory activities were under pH 4.5-5 and pH=7. more discussion is required

Author Response

The authors are grateful to the reviewers for their  comments to the manuscript, we present a new version of the manuscript that includes all the received  recommendations:

Line 90: Provide the accession number of the isolates.

Answer: 16S rRNA sequence obtained was deposited in GenBank (NCBI) MZ265376. Line 392. The whole genome of P. pentosaceus 1101 was deposited in the DDBJ/ENA/GeneBank under the accession number JAOAND000000000. Line 456.

Lines 92-94: how did the authors confirm the efficiency of the preservation and reactivation methods? reference is required.

Answer: the reference [20] was added in the text: Najjari, A.; Ouzari, H.; Boudabous, A.; & Zagorec, M. Method for reliable isolation of Lactobacillus sakei strains originating from Tunisian seafood and meat products. Int. J. Food Microbiol. 2008121(3), 342-351. Line 109.

Line 107: PCR conditions should be detailed

Answer: The employed PCR conditions were initial denaturation (1 cycle of 94 °C for 2 min), denaturalization (30 cycles of 94 °C for 0.5 min), alignment (30 cycles at 49 °C for 0.5 min), extension (30 cycles of 72 °C for 1 min) and final extension (1 cycle of 72 °C for 1 min). Lines 124-127.

Lines 195-199: according to the authors, a method by Khan [26] was employed with some modification. Nothing is mentioned about this adjustment in lines 195-199.

Answer: This expression was corrected in the text, as the used technique was the same with variation in the  growing conditions. Line 243.

Line 208: Clarify how the authors obtained swine bile.

Answer: Bile extract porcine was purchased form Sigma-Aldrich, MA, USA. Line 253.

Lines 351-354: Does data meet the assumption of homogeneity of variances? Clarify if the authors run a homogeneity test.

Answer: : Obtained data meet the assumption of homogeneity of variances. Line 438.

Lines 395-396: the best inhibitory activities were under pH 4.5-5 and pH=7. more discussion is required.

Answer: The efficacy of these antimicrobial peptides may depend on several factors including the pH, the presence of endogenous proteolytic enzymes, the membrane potential, and the lipid composition, that ultimately induce changes in the antimicrobial activity and their specificity. For instance, bacteriocins are stable in a wide range of pH with high activity at neutral and basic pH, but they are inactivated by endogenous proteases [48].Thus the ob-served higher antimicrobial activity at pH 5.5 and 7 for P. pentosaceus 1101 coincides with a decrement of its proteolytic activity. Lines 538-544